# OVRD: Open-Vocabulary Relation DINO with Text-guided Salient Query Selection

## Abstract

Open-Vocabulary Detection (OVD) trains on base categories and generalizes to novel categories with the aid of text embeddings from Vision-Language Models (VLMs). However, existing methods are insufficient in utilizing semantic cues from the text embeddings to guide visual perception, which hinders the performance of zero-shot object detection. In this paper, we propose OVRD, an **O**pen-**V**ocabulary **R**elation **D**INO with text-guided salient query selections. Specifically, we introduce text-guided salient query selection to choose image features most relevant to the text embeddings, along with their corresponding reference points and masks, thereby providing additional semantic cues for guiding visual perception. Building upon this, the salient reference points are used to recover the relative spatial structure of the selected features, enhancing positional awareness in the salient transformer decoder. Moreover, to fully leverage both the semantic cues and the recovered spatial structure, we develop a self-attention model of semantic relationships to model sparse semantic relations in OVD scenarios to further guide visual perception. We evaluate OVRD on public benchmarks in a zero-shot setting, achieving 37.0 AP on LVIS Minival, which performs favorably against the state-of-the-art methods. The code is available at https://anonymous.4open.science/r/OVRD.

## 1 Introduction

Traditional object detection is a fundamental task in computer vision. Numerous works Girshick (2015); Ren et al. (2015); Redmon et al. (2016); Carion et al. (2020); Zhang et al. (2023) accomplish promising detection performance. However, these methods are typically trained on datasets with closed-set categories, and thus struggle to recognize unseen categories.

To overcome this challenge, Open-Vocabulary Detection (OVD) Wu et al. (2024) is designed to detect both base and novel categories based on closed-set base categories. Recent pre-trained vision-language models (VLMs) Radford et al. (2021); Li et al. (2021) have demonstrated remarkable zero-shot capabilities, attributed to their training on massive and diverse image-text pairs. These models align image features and text embeddings in a shared embedding space, providing valuable priors for OVD. Early OVD methods, leveraging techniques, such as knowledge distillation Gu et al. (2022); Li et al. (2023); Ma et al. (2022); Wang et al. (2023) and region-text pre-training Zhong et al. (2022); Kim et al. (2024), compare image features and text embeddings mainly in the stage of classification, neglecting multi-modal fusion in the feature learning stage. In contrast, recent methods Cheng et al. (2024); Du et al. (2024); Wang et al. (2024) explicitly integrate multi-modal fusion into the model architecture. However, such fusion is still insufficient in leveraging the semantic cues to guide visual perception. Thus, we enhance it with a text-guided mechanism to improve zero-shot performance.

A text-guided mechanism provides additional semantic cues, which can be more effectively leveraged when combined with semantic relation modeling to improve object detection performance Yang et al. (2018); Xu et al. (2019); Hao et al. (2023). In this paper, we explicitly model semantic relations in open-vocabulary scenarios to fully exploit these semantic cues and guide visual perception. Figure 1 shows the key idea of the semantic relation modeling. We select an image[1]

---

[1]Figure 1(a) which is the ground-truth image from COCO, and Figure 1(b) which is the predicted image from our model.

Figure 1: **Demonstration of our semantic relation modeling in open-vocabulary scenarios.** (a) Given an image in the closed dataset which contains a base category "train", (b) the model detects additional novel objects in open-vocabulary scenarios, including "person", "street sign" and "traffic light". (c) We then capture the symmetric and fully-connected semantic relations with the aid of text-aware soft-mapping. (d) Finally, we model the directional and sparse relations to guide multi-modal fusion and improve zero-shot detection performance.

from the closed COCO Lin et al. (2014) dataset, containing a base category "train" as in Figure 1(a). In open-vocabulary scenarios, as shown in Figure 1(b), the model detects novel categories "person", "street sign", and "traffic light" in addition to the base category "train". Detecting more categories beyond the base classes enables a richer visual perception of the scene and captures more comprehensive semantic relations. Nevertheless, open-vocabulary settings introduce long-tailed and ambiguous category distributions, making accurate detection more challenging. To address this issue, we employ text-aware soft mapping to capture semantic embeddings of categories, which are then used to capture the symmetric and fully-connected semantic relations as in Figure 1(c). It is well-known that many relations are inherently asymmetric and directional, such as 'train → follows → traffic light' versus 'traffic light → guides → train'. Moreover, fully-connected relations among numerous objects can introduce redundancy and noise, which may hinder effective relation modeling. Therefore, we model the asymmetric and sparse relations to capture directionality and retain only the most informative relations, as illustrated in Figure 1(d). Modeling these semantic relations in open-vocabulary scenarios enhances visual perception and improves zero-shot detection performance.

In this paper, we propose Open Vocabulary Relation DINO (OVRD), a method that further enhances multi-modal fusion and explores relation modeling in open-vocabulary scenarios. Specifically, OVRD follows the standard DINO Zhang et al. (2023) architecture and leverages a pre-trained CLIP Radford et al. (2021) text encoder to extract text embeddings. We explore Text-guided Salient Query Selection (TSQS) to initialize queries in decoder and select text-relevant image features, reference points and masks utilized in salient multi-head attention to strengthen multi-modal fusion, providing additional semantic cues. However, this module discards the original spatial structure of encoder features, making it difficult to apply absolute sinusoidal positional embeddings. Thus, we leverage vision rotary positional embeddings to recover relative spatial structure and enhance positional awareness. Furthermore, we introduce Semantic Relation Self-Attention (SRSA) to fully utilize the semantic and spatial cues to better guide visual perception, which models semantic relations through text-aware soft mapping, while applying directionality to capture asymmetric relations and sparsification to avoid the disturbance from redundant relations. We train OVRD on large-scale datasets and evaluate the zero-shot performance on public OVD benchmarks.

Our main contributions are summarized as follows:

- We propose OVRD, an open-vocabulary detection model designed to improve zero-shot performance for real-world applications.
- We conduct Text-guided Salient Query Selection (TSQS) to select text-relevant features, reference points, and masks and improve text-guided visual perception.
- We introduce Semantic Relation Self-Attention (SRSA) to model sparse and directional semantic relations among object queries in open-vocabulary scenarios.
- OVRD is pre-trained on large-scale datasets and evaluated in a zero-shot setting, which achieves 29.6 AP on LVIS Val, surpassing OV-DINO by +2.7 AP. Ablation studies demonstrate our contributions to detection performance.

## 2 RELATED WORKS

### 2.1 DETECTION TRANSFORMERS

OVRD is built upon DINO Zhang et al. (2023), a DETR-like Transformer-based detection model. DETR Zhu et al. (2021) is inspired by the success of Transformers Vaswani et al. (2017), where features are enhanced by a Transformer encoder and static query embeddings are decoded without interaction with encoder features. DN-DETR Li et al. (2022a) adopts the same query selection method as DETR but feeds ground-truth bounding boxes with added noise into the decoder, leading to faster convergence. Deformable DETR Zhu et al. (2021) introduces deformable attention to accelerate convergence and reference boxes initialization through Top-K selection from encoder feature. Efficient DETR Yao et al. (2021) selects Top-K features based on classification score. DINO Zhang et al. (2023) further improves query selection based on the aforementioned methods and denoising techniques Li et al. (2022a), achieving strong performance. These traditional object detectors are trained on closed-set datasets with limited, pre-defined categories, therefore struggling to generalize to novel categories.

### 2.2 OPEN VOCABULARY OBJECT DETECTION

Open-Vocabulary Object Detection (OVD) Zareian et al. (2021) aims to detect both seen (base) and unseen (novel) categories by learning from seen categories, which differs from traditional object detection. Early approaches distill knowledge from pre-trained VLMs into object detectors Gu et al. (2022); Li et al. (2023); Ma et al. (2022); Wang et al. (2023). For instance, ViLD Gu et al. (2022) distills from teacher VLMs to compute image embeddings and text embeddings of regions. DK-DETR Li et al. (2023) introduces semantic and relational distillation schemes based on auxiliary queries to extract knowledge from VLMs. Although these distillation-based approaches are straightforward, their detection and generalization capabilities are inherently constrained by the teacher models. As VLMs are image-text pre-training, several methods propose to implement region-text pre-training Zhong et al. (2022); Kim et al. (2024) to fit detection task, but also lack multi-modal fusion. Recent methods pay more attention to the alignment and fusion of multi-modal features. YOLO-World Cheng et al. (2024) injects text features into image features through max-sigmoid attention to enhance multi-modal feature fusion. Several DETR-like models leverage text features to select queries and guide detection. Grounding-DINO Liu et al. (2024b) fuses image and text features via cross-attention in both Transformer encoder and decoder. OV-DINO Wang et al. (2024) utilizes text-aware object embeddings for query selection and introduces gated cross-attention in the decoder to improve multi-modal alignment. Our method further enhances multi-modal fusion and text-guided visual perception by leveraging text-guided salient query selection and semantic relation self-attention.

### 2.3 RELATION MODELING IN OBJECT DETECTION

The effectiveness of relation modeling between objects has been well demonstrated in object detection. Many image recognition methods Zhao et al. (2021); Chen et al. (2019) and region-based detectors Xu et al. (2019); Chen et al. (2021) compute correlation matrices and utilize Graph Convolutional Networks (GCNs) to model relational features. However, few works have explored relation modeling within Transformer-based detectors. Relation-DETR Hou et al. (2024) focuses on modeling explicit positional relationships by extracting geometric features of bounding boxes from each decoder layer, while leaving semantic relation modeling underexplored. Relation-enhanced DETR Hao et al. (2023) learns class correlations through a trainable relation matrix, which fails to generalize to novel categories in open-vocabulary settings due to its reliance on fixed class labels. Our method explores semantic relations in open-vocabulary scenarios through text-aware soft mapping, and models directional relations, while implementing sparsification to avoid interference from redundant and irrelevant connections.

## 3 METHODS

In this section, we first overview our proposed OVRD (Section 3.1), then introduce the Text-guided Salient Query Selection (Section 3.2), followed by the Positional Awareness Enhancement (Section 3.3), and finally the Semantic Relation Self Attention (Section 3.4).

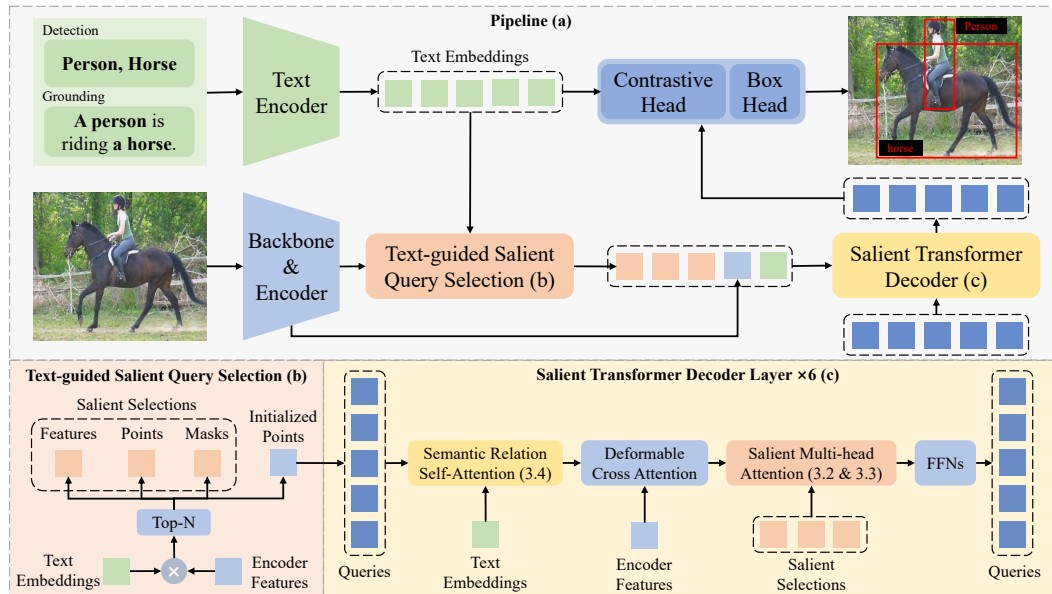

Figure 2: **Overall architecture of OVRD.** (a) OVRD builds upon DINO, where image features are extracted by the backbone and enhanced by the encoder. The text encoder receives detection and grounding texts to generate text embeddings. (b) Salient image features, along with their corresponding reference points and masks, are selected under the guidance of text embeddings. The initialized reference points are simultaneously selected according to these salient indices to initialize the learnable queries. (c) Salient Transformer Decoder iteratively refines the queries and predicts labels and bounding boxes via the contrastive heads and box heads.

## 3.1 OVERVIEW

The overall architecture of the proposed OVRD is illustrated in Figure 2 (a), which is basically built upon DINO Zhang et al. (2023). Given an image $I \in \mathbb{R}^{H \times W \times 3}$, multi-scale features are first extracted by the backbone and then flattened, while producing the image masks $M \in \{0, 1\}^{N_{\text{token}}}$ to indicate valid tokens and mask out the padded ones, where $(H \times W)$ is the original resolution of the image and $N_{\text{token}}$ denotes the number of flattened image tokens. The flattened image features, along with the positional embeddings, are fed into the transformer encoder to generate the encoder features $E_{\text{enc}} \in \mathbb{R}^{N_{\text{token}} \times D_I}$ and reference points $R_{\text{enc}} \in \mathbb{R}^{N_{\text{token}} \times 4}$, where $D_I$ denotes the dimension of image features.

In OVD, the processing and utilization of text is a key distinction from closed-set detection. Labels $T \in \mathbb{R}^C$ from detection and grounding datasets are first preprocessed, where $C$ is the number of nouns. Specifically, categories with single nouns in detection datasets are formatted as "a photo of a {}.". For grounding datasets, noun phrases are extracted from captions and formatted in the same way. Text encoder receives the processed texts and generate text embeddings $E_T \in \mathbb{R}^{C \times D_T}$, where $D_T$ denotes the dimension of text embeddings.

The image features $E_s \in \mathbb{R}^{N_Q \times D_I}$ most relevant to the text embeddings were selected in text-aware salient query selection (Figure 2 (b)), along with their reference points $R_s \in \mathbb{R}^{N_Q \times 4}$ and masks $M_s \in \{0, 1\}^{N_Q}$, to improve multi-modal fusion, where $N_Q$ is the number of queries. The initialized reference points of learnable queries are simultaneously selected according to these salient indices.

The initialized learnable queries $Q \in \mathbb{R}^{N_Q \times D_I}$ are fed into the salient transformer decoder (Figure 2 (c)) and first enhanced by semantic relation self attention (Figure 3) to focus on semantic relations. They are then refined by deformable cross-attention where the encoder features serve as the memories. Additionally, the queries are updated via a salient multi-head cross-attention mechanism, in which the salient memories emphasize the most text-relevant visual cues. However, these salient selections discard the original spatial structure of encoder features. Thus, we employ vision rotary position embeddings for both the selected salient memories and queries in salient multi-head atten-

tion to better recover relative spatial structure. Finally, the updated queries are passed through feed-forward networks (FFNs) and the subsequent contrastive head and box head to produce the labels and bounding box predictions, where the contrastive head is implemented as a cosine-similarity–based classifier. Specifically, the text embeddings are first projected to the same dimension as the visual queries using a single linear layer. Both visual queries and projected text features are then L2-normalized, and compute their cosine similarity to produce the class logits, which are converted into probability distributions through a softmax operation.

## 3.2 TEXT-GUIDED SALIENT QUERY SELECTION

Text-guided Salient Query Selection identifies the image features most relevant to the text embeddings. Along with these text-related salient image features, the corresponding reference points and masks are also selected as shown in Figure 2 (b). These salient selections, including the salient features, reference points, and masks, are then sent into salient multi-head attention, highlighting the image features most relevant to the input text. In this way, this module provides additional semantic cues and helps the queries focus on the most semantically relevant regions, improving the guidance of visual perception by text embeddings.

**Review of Text-guided query selection.** Query selections in DETR-series are continuously evolving as introduced in Section 2.1. To better guide open-vocabulary detection, Grounding DINO Liu et al. (2024b) is inspired by DINO Zhang et al. (2023) to propose text-guided query selection[2], integrating its query selection module with text embeddings to select the text-relevant encoder features. OV-DINO Wang et al. (2024) further improves text-guided query selection[3] by selecting text-related salient features[4] $E_s$, which served as keys in the added salient multi-head attention. Specifically, the top $N_Q$ encoder features are selected under the guidance of text embeddings, as proposed in OV-DINO Wang et al. (2024):

$$E_s, K_s = \text{Top}_{N_Q}(\mathcal{T}_{\text{CLS}}(\|E_{\text{enc}}\|_2, \|E_{PT}\|_2)), \tag{1}$$

where $K_s \in \mathbb{R}^{N_Q}$ denote the salient indices, and $\mathcal{T}_{\text{CLS}}$ denotes the contrastive classifier, which is implemented as the classification branch of the Contrastive Head introduced in section 3.1. The initialized reference points of queries are simultaneously selected according to these salient indices.

**Text-guided Salient Query Selection.** As described in overview (Section 3.1), multi-scale image features are first flattened and sent into the transformer encoder with absolute positional embeddings, which provides implicit global positional information to the encoder features. Feature masks are calculated through multi-scale image features to prevent attention on invalid or padding tokens, ensuring that the model focuses only on meaningful spatial features.

However, implementing the Top-K operation solely on encoder features without updating the corresponding reference points causes salient features to lose their associated positional information. Similarly, neglecting the corresponding masks diminishes the attention's focus on actual objects.

Therefore, we adopt text-guided salient query selection to obtain complete salient selections, including the salient features with the attached salient reference points and masks. Specifically, salient features are obtained as in Equation 1, while the reference points $R_s$ and masks $M_s$ are selected as:

$$R_s = \{R_{\text{enc},k} \mid k \in K_s\}, \quad M_s = \{M_k \mid k \in K_s\}. \tag{2}$$

Note that both the original masks $M$ and the salient masks $M_s$ are calculated during training only. We give a visualization of the salient tokens over text-guided relevance heatmap in Appendix A.3 for better understanding of the proposed TSQS module.

## 3.3 POSITIONAL AWARENESS ENHANCEMENT

In DETR-series models, positional embeddings (PEs) explicitly encode spatial order in Transformer-based models, allowing them to distinguish element positions and improve structural relation modeling and contextual understanding.

---

[2]Originally called *Language-Guided Query Selection* in Grounding DINO.

[3]Originally called *Language-Aware Selective Fusion* in OV-DINO.

[4]Originally called *Object Embeddings* in OV-DINO.

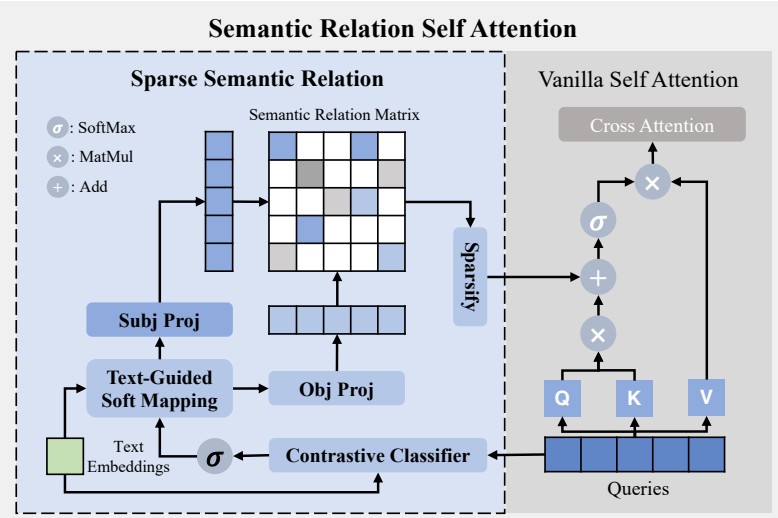

Figure 3: **Detailed illustration of the Semantic Relation Self Attention.** This module first performs contrastive classification of queries with text embeddings at each layer, followed by operating softmax to obtain the probability distribution over labels. The result distribution is then transformed into high-level semantics through text-aware soft mapping. These semantic representations are subsequently projected as subjects and objects, emphasizing their different roles in directional relations. The outputs are multiplied to generate a high-level semantic relation matrix. Finally, the matrix is sparsified and integrated into the self-attention mechanism.

**Positional Embeddings in Salient Transformer Decoder.**    In the salient transformer decoder, the original reference points calculated in text-guided salient query selection (Section 3.2) are used to generate the initialized PEs of the queries. Sinusoidal PEs are commonly used in DETR-series models to calculate absolute position information between tokens. Queries in deformable cross attention share the same PEs, while the reference points are explicitly utilized to compute their spatial locations. However, selecting the most text-relevant features from the encoder discards the original grid-based spatial structure. This makes it challenging for the additional salient multi-head attention to correctly infer the positions of the selected feature tokens. Even though the salient reference points are simultaneously chosen to calculate the absolute PEs, the effect is still limited due to the loss of the global spatial continuity of the original dense grid.

**Vision Rotary Positional Embeddings.**    To mitigate these issues and enhance the positional awareness, we leverage vision rotary positional embeddings (RoPE) Heo et al. (2024) for queries and keys in salient multi-head attention, while keeping the other PEs in salient transformer decoder unchanged. Vision RoPE injects relative positions into the query and key vectors through rotation, allowing the attention weights to naturally reflect relative positions between tokens. Specifically, RoPE Su et al. (2024) in language model utilizes the multiplication of Euler's formula ($e^{i\theta}$) to inject relative positions. Vision RoPE extends this idea by considering axial frequencies, expanding 1D RoPE to horizontal and vertical axes, and further implementing mixed learnable frequencies to capture relations along diagonal directions. By extracting relative positional information from salient reference points, vision RoPE helps recover the spatial structure and provides relative positional cues, enabling attention to reason about positions even when features are sparsely selected.

## 3.4 SEMANTIC RELATION SELF ATTENTION

Modeling semantic relations benefits object detection, which has been widely demonstrated. Semantic Relation Self-Attention (SRSA), as depicted in Figure 3, guides the self-attention mechanism to focus on semantic relations in open-vocabulary scenarios using text embeddings. This module fully utilizes the semantic cues from TSQS (Section 3.2) and relative spatial cues from vision RoPE (Section 3.3) to further enhance multi-modal fusion.

**Label Distribution via Contrastive Classifier.** This module first obtains probability distribution over labels through a contrastive classifier. Specifically, given queries $Q$ in each decoder layer and text embeddings $E_T$ from text encoder with its projector $W_T \in \mathbb{R}^{D_I \times D_T}$, we first compute the projected text embeddings $E_{PT} = E_T W_T^\top, E_{PT} \in \mathbb{R}^{C \times D_I}$. The probability distribution $P \in \mathbb{R}^{N_Q \times C}$ is then obtained by the contrastive classifier $\mathcal{T}_{\text{CLS}}$ after a softmax operation $\sigma$:

$$P = \sigma(\mathcal{T}_{\text{CLS}}(\|Q\|_2, \|E_{PT}\|_2)). \tag{3}$$

**Text-aware Soft Mapping.** Prior methods Yang et al. (2018); Zhao et al. (2021); Hao et al. (2023) compute semantic relations based on weights of linear classifier, since they are trained on closed datasets with limited categories, which allows the classifier to easily capture distinct semantics from different labels. However, in open-vocabulary scenarios, labels are not fixed during training, and evaluation occurs on datasets with long-tail and ambiguous categories. These challenges necessitate robust characterization of high-level semantic embeddings for category prototypes, motivating the Text-aware Soft Mapping. Specifically, the probability distribution is multiplied with projected text embeddings to produce the high-level semantic embeddings $E_{\text{hls}} \in \mathbb{R}^{N_Q \times D_I}$:

$$E_{\text{hls}} = P E_{PT}. \tag{4}$$

In this way, the classification probability space is mapped into the semantic space, where the probability distribution acts as weights over the text embeddings, enhancing semantic awareness of queries and improving the alignment between queries and the semantic space.

**Directional Relations Modeling.** As mentioned in introduction (Section 1), semantic relations are asymmetric and directional. To capture such relations, we employ two separate MLPs with identical structures, denoted as $\text{MLP}_{\text{subj}} : \mathbb{R}^{D_I} \to \mathbb{R}^{D_R}$ and $\text{MLP}_{\text{obj}} : \mathbb{R}^{D_I} \to \mathbb{R}^{D_R}$, on the obtained high-level semantic embeddings, where $D_R$ represents the dimension of the relations and is set to 64 by default. Each MLP contains two linear layers with an activation function in between. This design helps select targets with directional semantics, enhancing both the flexibility and accuracy of relation modeling. The semantic relation matrix $\text{SR} \in \mathbb{R}^{N_Q \times N_Q}$ can be calculated as:

$$\text{SR} = \text{MLP}_{\text{subj}}(E_{\text{hls}})\text{MLP}_{\text{obj}}(E_{\text{hls}})^\top. \tag{5}$$

**Semantic Relation Sparsification.** Numerous categories in open-vocabulary scenarios result in large fully-connected relation matrices, which inevitably include redundant connections and thereby hinder accurate relation modeling. To address this, we sparsify the semantic relation matrix by selecting semantic relations with higher scores for each query. Specifically, the sparsification follows the under expression:

$$\text{SSR}_{i,j} = \text{SR}_{i,j} \cdot \mathbf{1}[\text{SR}_{i,j} \in \text{Top}_{SN}(\text{SR}:,j)], \tag{6}$$

where $\text{SSR} \in \mathbb{R}^{N_Q \times N_Q}$ denotes the sparse semantic relation, $\mathbf{1}(\cdot)$ equals 1 when the condition holds and 0 otherwise, $\text{Top}_{\text{SN}}$ picks the top SN elements and SN is a sparse number controlling how many elements are retained, which is set to 32 by default.

**Integration into Self-Attention.** Finally, the sparse semantic relation is integrated into the vanilla self-attention mechanism as follows:

$$\text{SRSA}(Q, E_T) = \sigma(\mathbf{SSR(Q, E_T)} + \frac{\text{Que}(Q)\text{Key}(Q)^\top}{\sqrt{D_I}})\text{Val}(Q). \tag{7}$$

This integration allows the module to model sparse semantic relations in open-vocabulary scenarios. By incorporating textual information, the self-attention mechanism is enhanced in capturing semantic relations and improving multi-modal fusion.

## 4 EXPERIMENTS

In this section, we demonstrate the effectiveness of OVRD, which is pre-trained on large-scale datasets and evaluated in a zero-shot setting. We introduce the datasets and the evaluation metric in Section 4.1, then the implementation details in Section 4.2. The main result and comparisons with other methods are then present in Section 4.3, and finally the ablation studies in Section 4.4.

Table 1: **Zero-shot evaluation on LVIS.** We evaluate OVRD for fixed AP on LVIS minival and LVIS val in a zero-shot setting and compare with other recent methods. AP for LVIS minival is the main metric. AP with subscripts r, c, and f denotes AP for rare, common and frequent categories, respectively. In the column of Datasets, O means Objects365v1, G is GoldG, VG is Visual Genome Krishna et al. (2017). † means the re-evaluated results, discussed in Appendix A.5.

| Model | Datasets | LVIS MiniVal | | | | LVIS Val | | | |
|---|---|---|---|---|---|---|---|---|---|
| | | **AP** | $AP_r$ | $AP_c$ | $AP_f$ | AP | $AP_r$ | $AP_c$ | $AP_f$ |
| GLIP-T(B) Li et al. (2022b) | O | 17.8 | 13.5 | 12.8 | 22.2 | 11.3 | 4.2 | 7.6 | 18.6 |
| G-DINO-T Liu et al. (2024b) | O,G | 25.6 | 14.4 | 19.6 | 32.2 | – | – | – | – |
| LAMI-DETR Du et al. (2024) | O,VG | 35.4 | **37.8** | – | – | – | – | – | – |
| YOLO-W-L Cheng et al. (2024) | O,G | 35.2 | 27.8 | 32.6 | 38.8 | 28.3 | 22.5 | 24.4 | 35.1 |
| YOLOE-v8-L Wang et al. (2025) | O,G | 35.9 | 33.2 | 34.8 | 37.3 | – | – | – | – |
| Open-Det Cao et al. (2025) | VG | 33.1 | 31.2 | 32.1 | 34.3 | – | – | – | – |
| OV-DINO[1]† Wang et al. (2024) | O | 21.2 | 7.9 | 16.6 | 27.7 | 16.5 | 6.8 | 12.4 | 25.3 |
| OV-DINO[2]† Wang et al. (2024) | O,G | 36.1 | 32.9 | **35.0** | 37.7 | 26.9 | **24.2** | **27.8** | 34.0 |
| OVRD-T[1] (Ours) | O | 28.1 | 23.0 | 26.2 | 30.8 | 22.4 | 17.8 | 19.5 | 27.6 |
| OVRD-L[1] (Ours) | O,G | **37.0** | 33.1 | 33.4 | **40.9** | **29.6** | 22.4 | 26.0 | **36.7** |

## 4.1 DATASETS AND METRIC

For fair comparison with existing methods, we pre-trained OVRD on large-scale datasets, including detection dataset Objects365v1 Shao et al. (2019) and grounding dataset GoldG Kamath et al. (2021) (GQA Hudson & Manning (2019) and Flickr30k Plummer et al. (2015)). We evaluate our method on the LVIS dataset Gupta et al. (2019) in a zero-shot setting and report Fixed AP Dave et al. (2021) on LVIS Minival and LVIS Val for comparison. We also evaluate OVRD on COCO2017 Val Lin et al. (2014) in a zero-shot setting and report mean AP for fair comparison. Details of datasets are provided in Appendix A.2.

## 4.2 IMPLEMENTATION DETAILS

We conduct the main experiments on 8 40G A100 GPUs with batch size 4 for each GPU. We provide two scales of the proposed OVRD. OVRD-T utilizes Swin-T Liu et al. (2021) as image backbone and is trained on Objects365v1 dataset for 12 epochs. OVRD-L utilizes Swin-L as image backbone and is trained on both Objects365v1 and GoldG datasets for 30 epochs. CLIP-B Radford et al. (2021) text encoder is implemented for both models, and we follow YOLO-UniOW Liu et al. (2024a) to utilize LoRA Hu et al. (2022) to fine-tune the text encoder. Following the settings in mm-grounding-dino Zhao et al. (2024) and other DINO-based OVD methods Wang et al. (2024); Du et al. (2024), we adopt the AdamW Loshchilov & Hutter (2019) optimizer with a weight decay of 1e-4. Base learning rate is 1e-4 for both the model and LoRA-fine-tuned text encoder, while it is $0.1\times$ the base learning rate to the image backbone. Moreover, we use a multi-step learning rate schedule. Learning rate in OVRD-T is reduced to 0.1 times base learning rate after 10 epochs, while in OVRD-L, the learning rate is reduced to 0.1 and 0.01 times base learning rate after 19 and 26 epochs, respectively. Detailed parameters are largely identical to the original DINO and are provided in Appendix A.6.

## 4.3 MAIN RESULTS

The main results are shown in Table 1 and the comparison with recent state-of-the-art methods are also provided. For the result on LVIS Minival, OVRD-T, trained only on the Objects365v1 for 12 epochs, achieves 28.1 AP, outperforming OV-DINO[1] by 6.9 AP and even surpassing Grounding-DINO-T, which is trained on both Objects365v1 and GoldG, by 2.5 AP. OVRD-L is trained on both Objects365v1 and GoldG datasets, and it achieves 37.0 AP, outperforming OV-DINO[2] by 0.9 AP. For the result on LVIS Val, OVRD-T achieves 22.4 AP, outperforming OV-DINO[1] by 5.9 AP, while OVRD-L is 29.6 AP, surpassing OV-DINO[2] by 2.7 AP. OVRDs achieve superior performance, presenting remarkable zero-shot abilities. Though OVRD-L has relatively poor $AP_r$ and $AP_c$ compared to OV-DINO[2], it achieves the better AP and $AP_f$, demonstrating strong overall recognition ability. Visualization results are shown in Appendix A.3.

Table 2: **Zero-shot evaluation on COCO2017 Val.** We directly evaluate pre-trained OVRD on COCO2017 Val in a zero-shot setting and compare with other recent methods. Mean AP is reported as the main metric, and $AP_{50}$ and $AP_{75}$ are also provided for reference.

| Model | Zero-shot COCO2017 Val | | |
|---|---|---|---|
| | **AP** | $AP_{50}$ | $AP_{75}$ |
| YOLO-World-S Cheng et al. (2024) | 37.6 | 52.3 | 40.7 |
| YOLO-World-M Cheng et al. (2024) | 42.8 | 58.3 | 46.4 |
| YOLO-World-L Cheng et al. (2024) | 44.4 | 59.8 | 48.3 |
| Open-Det Cao et al. (2025) | 35.8 | – | – |
| G-DINO-T Liu et al. (2024b) | 48.1 | – | – |
| OV-DINO[1] Wang et al. (2024) | 49.5 | – | – |
| OVRD-T (Ours) | 44.9 | 60.2 | 49.1 |
| OV-DINO[2] Wang et al. (2024) | 50.6 | – | – |
| OVRD-L (Ours) | **51.2** | **66.9** | **56.2** |

Table 3: **Ablations on OVRD Components.** OVRD improves detection performance through the salient selections, the use of RoPE, and the semantic relations. Numbers in parentheses denote the gain compared to the previous row / baseline.

| Methods | AP | $AP_r$ | $AP_c$ | $AP_f$ |
|---|---|---|---|---|
| Baseline | 19.7 | 16.9 | 17.7 | 22.0 |
| **Cumulative Effect** | | | | |
| + Text-guided Salient Query Selections (3.2) | 21.3 (+1.6 / +1.6) | 14.4 | 17.9 | 25.6 |
| + Position Awareness Enhancement (3.3) | 22.0 (+0.7 / +2.3) | 18.1 | 20.0 | 24.6 |
| + Semantic Relation Self-Attention (3.4) | **23.4** (+1.4 / +4.2) | **21.2** | **21.3** | **25.6** |
| **Independent Effect (added to Baseline only)** | | | | |
| Baseline + Semantic Relation Self-Attention (3.4) | 21.8 (+2.1) | 18.0 | 19.8 | 24.2 |

We also evaluate OVRD on COCO Lin et al. (2014) in a zero-shot setting after pre-training and compare with other methods in Table 2. Mean AP is reported as the main metric, and $AP_{50}$ and $AP_{75}$ are also provided for reference. OVRD-L achieves 51.2 AP, surpassing OV-DINO[2] by 0.6 AP, demonstrating the effectiveness of our proposed method. OVRD-T also achieves 44.9 AP, which is also competitive among YOLO-World and Open-Det. However, OVRD-T is inferior to Grounding-DINO-T and OV-DINO[1] on COCO. This is because OVRD-T is only trained on Objects365v1 for 12 epochs, while Grounding-DINO-T and OV-DINO[1] are trained on for a longer period.

## 4.4 ABLATION STUDIES

We conduct ablation studies to analyze OVRD. We randomly sample 20% of the whole training dataset (OG) with a fixed random seed to reduce the training cost while keeping relatively sufficient and diverse training data. All ablation studies are conducted on 8 V100 GPUs under OVRD-T settings. The results of adding semantic relations (Table 3), 32 of the sparse number (Table 4) and RoPE of the positional embeddings (Table 5) are based on the same default model.

**Ablations on OVRD components.** Table 3 demonstrates the impact of each component introduced into our OVRD. The reproduced OV-DINO is used as the baseline with the same settings as OVRD-T, where its original BERT text encoder is replaced by CLIP with LoRA for fair comparison, achieving 19.7 AP. We first evaluate the cumulative results of each component. We set the positional embeddings of keys to Sinusoidal PE calculated by reference points from text-guided query salient selection and achieve remarkable improvement (+ 1.6 AP). The implementation of RoPE instead of Sinusoidal PE enhances position awareness (+ 0.7 AP). OVRD computes the semantic relation in self-attention, which achieves 23.4 AP and totally + 4.2 AP improvement compared to the baseline in the random selected datasets. We also evaluate the result of independent effect, which adds semantic relation self-attention to the baseline only, achieving + 2.1 AP improvement.

Table 4: **Ablations on Sparse Number.** Different sparse number of semantic relation are evaluated.

| Sparse Number | AP | $AP_r$ | $AP_c$ | $AP_f$ |
|---|---|---|---|---|
| 0 (w/o Sparse) | 21.7 (- 1.7) | 15.1 | 19.0 | 25.3 |
| 16 | 22.4 (- 1.0) | 19.3 | 21.1 | 24.0 |
| 32 (default) | **23.4** | **21.2** | **21.3** | **25.6** |
| 64 | 23.1 (- 0.3) | 19.0 | 21.3 | 25.6 |
| 128 | 21.9 (- 1.5) | 19.8 | 19.7 | 24.2 |
| 256 | 21.7 (- 1.7) | 18.3 | 20.0 | 23.7 |
| 512 | 21.5 (- 1.9) | 18.9 | 19.0 | 24.2 |

**Ablations on Sparse Number.** Table 4 presents the effectiveness of different number to sparsify the semantic relations. Since the total number of queries is about 1100 (900 initialized and 200 denoising), we set the maximum number of sparse queries to 512, which is nearly half of all queries. When the sparse number is 64, the result is close to our default setting. When the sparse number is small (16), some meaningful relations may be overlooked, leading to performance degradation. However, when the sparse number is 128 or higher, or when sparsification is not applied (0), noisy and redundant connections may disrupt the self-attention, resulting in decreased performance.

Table 5: **Ablations on Positional Embeddings.** Different positional embeddings are evaluated.

| Positional Embeddings (PE) | AP | $AP_r$ | $AP_c$ | $AP_f$ |
|---|---|---|---|---|
| w/o PE | 22.0 (-1.4) | 15.0 | 20.8 | 24.2 |
| Sinusoidal PE | 22.3 (- 1.1) | 11.8 | 19.3 | **26.9** |
| Learnable PE | 23.2 (- 0.2) | 20.3 | **21.6** | 25.1 |
| RoPE (default) | **23.4** | **21.2** | 21.3 | 25.6 |

**Ablations on Positional Embeddings.** Table 5 shows the impact of different positional embeddings (PE). We first evaluate the performance without PE, where only salient masks are used, which results in a drop in detection performance (- 1.4 AP). Using Sinusoidal PE slightly improves performance compared to not using any PE (+ 0.3 AP). However, it still lags behind RoPE (- 1.1 AP), mainly due to the loss of original spatial information caused by text-guided salient query selections. We also evaluate Learnable PE, but it still fails to match the performance of RoPE (- 0.2 AP).

## 5 CONCLUSION

In this paper, we present OVRD, an approach to improve open-vocabulary detection performance. We introduce text-guided salient query selection to enhance multi-modal fusion and further improve positional awareness using salient reference points. Moreover, we explore semantic relation modeling in open-vocabulary scenarios and integrate it into self-attention to strengthen text-guided visual perception. Evaluations on LVIS demonstrate that OVRD achieves remarkable performance in open-vocabulary detection.

## 6 REPRODUCIBILITY STATEMENT

We are committed to ensuring the reproducibility of our work. The detailed model architecture is presented in Section 3. Experimental settings, including datasets, implementation details, and evaluation metrics, are provided in Section 4. We also provide appendix in Section A with additional implementation details, including detailed dataset information, parameter settings, visualization results, and the re-evaluation of compared methods. Codes and pre-trained weights are made available through the anonymous link in the abstract to facilitate reproduction of our results.

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

## A  APPENDIX

### A.1  THE USE OF LARGE LANGUAGE MODELS (LLMS)

We used LLMs, mostly ChatGPT, to assist with language polishing and code development, while all research ideas and results remain solely the authors' work.

## A.2 DETAIL INFORMATION OF DATASETS

We pre-train our OVRD on detection dataset Objects365v1 Shao et al. (2019) and grounding dataset GoldG Kamath et al. (2021) and evaluate on LVIS Gupta et al. (2019). Datasets information is listed in table 6.

Table 6: **Brief Statistics of used datasets**

| Dataset | Type | Classes/Texts | Images | Anno. |
|---------|------|---------------|--------|-------|
| Objects365v1 | Detection | 365 | 609K | 9621K |
| GQA | Grounding | 387K | 621K | 3681K |
| Flickr30k | Grounding | 94K | 149K | 641K |
| COCO2017 Val | Detection | 80 | 5K | – |
| LVIS Val | Detection | 1203 | 20K | – |
| LVIS minival | Detection | 1203 | 5K | – |

Objects365 is a large-scale detection dataset with 365 classes. Objects365v1 has 609K images and nearly 100M annotations. Objects365v2 is a much larger dataset with 2M images and over 300M annotations. We use Objects365v1 to train our model.

The grounding dataset GoldG consists of GQA Hudson & Manning (2019) and Flickr30k Plummer et al. (2015) and excludes images from COCO to obey the zero-shot setting. GQA is for visual question answering and Flickr30k is for sentence-based image description.

The COCO 2017 Validation set (COCO2017 Val) Lin et al. (2014)is a widely used benchmark for object detection. It contains 5,000 images selected from the COCO dataset and provides high-quality annotations for 80 object categories COCO2017 Val is primarily used for model validation and performance comparison, as it offers a balanced and diverse set of everyday scenes while remaining small enough for efficient evaluation.

LVIS (V1 in our evaluation) is based on COCO Lin et al. (2014) with the same images but different annotations. LVIS has long-tail categories and some of them have only a few examples, which makes it a hard dataset for detection. LVIS Minival has the same images with COCO2017 Val and is the main evaluated dataset.

## A.3 VISUALIZATION OF RESULTS

The visualization results of OVRD-L on LVIS Minival is presented in Figure 4, and we also compare with OV-DINO. From the visualization results, we can see that OVRD-L can detect more objects with higher confidence, especially for small objects and densely crowded scenes. This demonstrates the effectiveness of our selected salient reference points and the use of vision RoPE, which together enable the model to better capture inter-object spatial relationships.

## A.4 VISUALIZATION OF SALIENT TOKENS OVER TEXT-GUIDED RELEVANCE HEATMAP

To better illustrate how the proposed TSQS module identifies text-relevant visual evidence, we visualize the text-guided saliency heatmap together with the Top-K selected salient features in Figure 5. For each image, we first compute the text-guided relevance score for every encoder token by measuring its similarity to the input text embeddings, producing a dense heatmap that highlights visually informative and text-relevant regions. We then apply our selection mechanism to extract the Top-K salient tokens, whose spatial locations are shown as points overlaid on the heatmap. As shown in the figure, the salient tokens selected by our TSQS module consistently fall within the most text-relevant regions. The density of selected points naturally increases around areas with strong semantic meaning, indicating that our method is able to accurately capture and prioritize informative visual evidence. This behavior demonstrates the effectiveness of our approach in identifying text-aligned regions, especially for challenging cases such as small objects or densely crowded scenes.

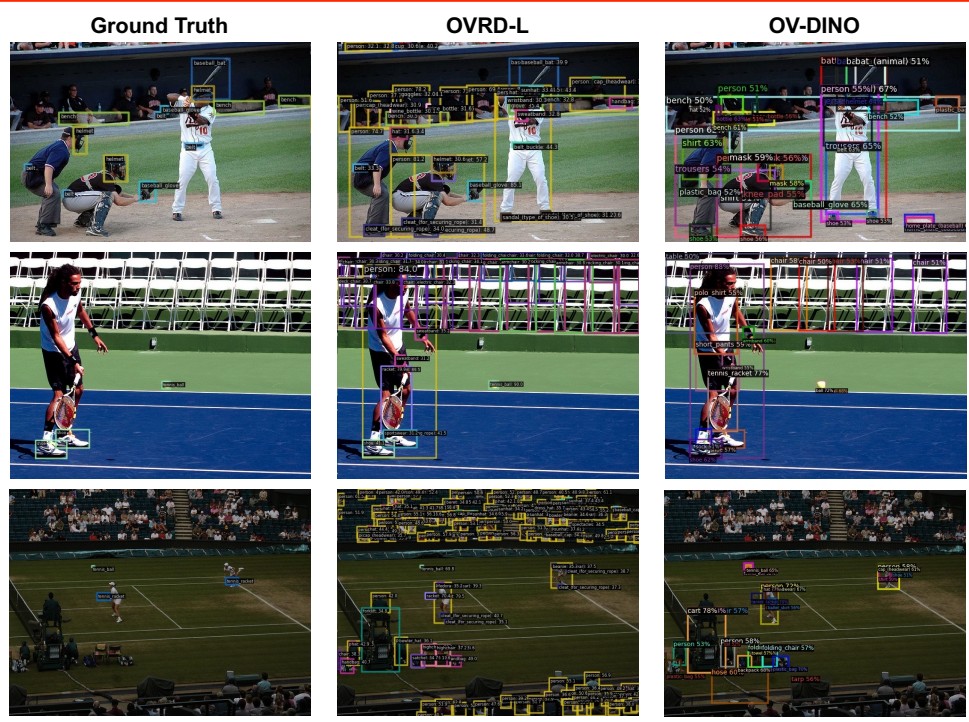

Figure 4: **Visualization results of OVRD-L and OV-DINO on LVIS Minival.** The left part of each couple of images is the ground-truth, and middle is from OVRD-L and the right is from OV-DINO form comparison.

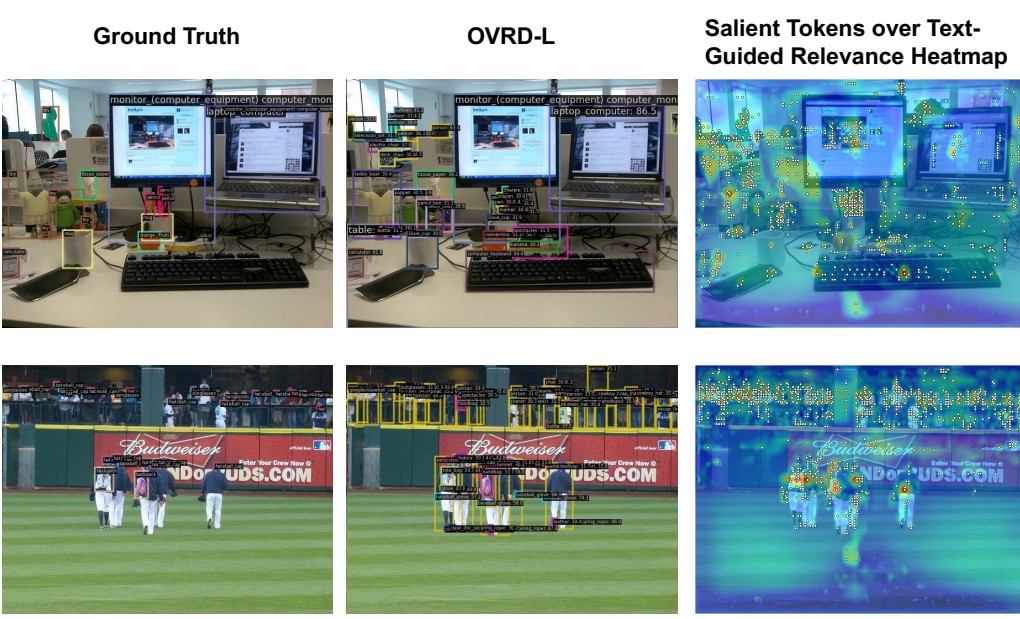

Figure 5: **Visualization of salient tokens over text-guided relevance Heatmap.** The left part of each couple of images is the ground-truth, and middle is the result from OVRD-L. The right part shows the heatmap generated by evaluating the relevance between image features and text embeddings, while the points indicate the selected salient tokens by TSQS.

Table 7: **Ablation of Multi-template and Chunk Dataset on LVIS MiniVal.**

| Model | LVIS MiniVal | | | |
|---|---|---|---|---|
| | **AP** | $AP_r$ | $AP_c$ | $AP_f$ |
| **Baseline (Ours-reported)** | | | | |
| OV-DINO[1] | 21.2 | 7.9 | 16.6 | 27.7 |
| OVRD-T | 28.1 | 23.0 | 26.2 | 30.8 |
| OV-DINO[2] | 36.1 | 32.9 | 35.0 | 37.7 |
| OVRD-L | 37.0 | 33.1 | 33.4 | 40.9 |
| **w/ Multi-template with pooling** | | | | |
| OV-DINO[1] | 21.2 (+0.0) | 7.9 (+0.0) | 16.8 | 27.4 |
| OVRD-T | 28.2 (+0.1) | 24.9 (+1.9) | 26.2 | 30.5 |
| OV-DINO[2] | 38.2 (+2.1) | 34.6 (+1.7) | 37.3 | 39.6 |
| OVRD-L | 38.9 (+1.9) | 34.9 (+1.8) | 35.5 | 42.3 |
| **w/ Chunk dataset** | | | | |
| OV-DINO[1] | 24.2 (+3.0) | 15.3 (+7.4) | 19.8 | 29.7 |
| OVRD-T | 31.0 (+2.9) | 31.0 (+8.0) | 29.6 | 32.2 |
| OV-DINO[2] | 37.2 (+0.9) | 33.1 (+0.2) | 36.3 | 38.7 |
| OVRD-L | 37.6 (+0.6) | 33.1 (+0.0) | 34.5 | 41.6 |
| **w/ Multi-template with pooling & Chunk dataset (OV-DINO-reported)** | | | | |
| OV-DINO[1] | 24.4 (+3.2) | 15.5 (+7.6) | 20.3 | 29.7 |
| OVRD-T | 31.1 (+3.1) | 31.3 (+8.3) | 30.1 | 32.4 |
| OV-DINO[2] | 39.4 (+3.3) | 32.0 (-0.9) | 38.7 | 41.3 |
| OVRD-L | 40.2 (+3.2) | 33.6 (+0.5) | 36.3 | 43.9 |

## A.5 RE-EVALUATING OV-DINO

We evaluate OV-DINO[1] and OV-DINO[2] using their provided code and weights. However, post-training tricks are applied in the original implementations, which we discard in our re-evaluation to ensure a fair comparison with our method and other methods Cheng et al. (2024); Du et al. (2024); Wang et al. (2025); Liu et al. (2024b); Cao et al. (2025).

We conducted ablation studies on these two tricks used in OV-DINO, and we also provide our results with these tricks in Table 7. We directly use the pretrained weights of OV-DINO[1] and OVRD-T without any modifications in models. From Table 7, we observe that these inference-time tricks significantly improve performance without modifying the model architecture. When using only multi-template with pooling, the overall AP barely changes, but OVRD achieves a notable +1.9 $AP_r$ improvement, indicating enhanced recognition of rare categories. When using only the chunk dataset, the gains become much more pronounced. OV-DINO improves by +3.0 AP, and OVRD-T improves by +2.9 AP, with especially large benefits for rare categories, OV-DINO achieves +7.4 $AP_r$, while OVRD-T reaches +8.0 $AP_r$. When combining both tricks, the improvements go even further than either trick alone.

We further evaluate the impact on the larger models trained with Objects365 and GoldG (i.e., OV-DINO[2] and OVRD-L). The corresponding results are also provided in Table 7. Although the absolute performance differs from the smaller models, the overall trends remain consistent across OV-DINO[2] and OVRD-L. When applying only multi-template pooling, both OV-DINO[2] and OVRD-L exhibit clear performance gains (+2.1 AP and +1.9 AP, respectively). These increases are even larger than those observed in the Objects365-only models, indicating that incorporating grounding-style annotations (GoldG) enhances the model's sensitivity to text embeddings. In contrast, the chunked label set brings smaller improvements for the larger models (+0.9 AP for OV-DINO[2] and +0.6 AP for OVRD-L), particularly for rare categories. This is expected, as additional grounding data already mitigates long-tail imbalance and improves recognition of rare classes, thereby reducing the marginal benefits of chunk-based label balancing. Overall, these results clearly demonstrate that both multi-template pooling and the chunk dataset significantly enhance zero-shot recognition while requiring no changes to the underlying model.

Table 8: **Parameters of OVRDs**

| Parameters | OVRD-T | OVRD-L |
|---|---|---|
| **Training Settings** | | |
| Batch size per GPU | 4 | 4 |
| Epochs | 12 | 30 |
| Datasets | O | O,G |
| Image Backbone | Swin-T | Swin-L |
| Text Encoder | CLIP-B+LoRA | CLIP-B+LORA |
| Text Format | a photo of a {}. | a photo of a {}. |
| Optimizer | AdamW | AdamW |
| Base Learning Rate (lr) | 1e-4 | 1e-4 |
| lr decay milestones ($\times$ ratio) | 10 ($\times$ 0.1) | 19 ($\times$ 0.1) 26($\times$ 0.01) |
| lr for image backbone | lr $\times$ 0.1 | lr $\times$ 0.1 |
| lr for text encoder | CLIP: 0, LoRA: lr | CLIP: 0, LoRA: lr |
| Weight Decay | 1e-4 | 1e-4 |
| Warmup iter | 1000 | 1000 |
| **Model Parameters** | | |
| Training Categories ($C$) | 80 | 80 |
| Image Feat Dim ($D_I$) | 256 | 256 |
| Text Embed Dim ($D_T$) | 512 | 512 |
| Num of queries ($N_Q$) | 900 | 900 |
| Sem Rel Dim ($D_R$) | 64 | 64 |
| Sparse Number ($SN$) | 32 | 32 |
| Enc Layers | 6 | 6 |
| Enc FFN Activation | SiLU | SiLU |
| Dec Layers | 6 | 6 |
| Num of Heads | 8 | 8 |
| Dec FFN Activation | SiLU | SiLU |
| **Loss Function** | | |
| Losses | Focal,L1,GIoU | Focal,L1,GIoU |
| Costs of Losses | 1,5,2 | 1,5,2 |
| Weights of Losses | 2,5,2 | 2,5,2 |

We then introduce the two tricks used in OV-DINO. Multi-template with pooling refers to applying multiple textual templates to the same category label during inference. These templates are wrapped around the category names and fed into the text encoder. After computing the text embeddings for all templates, pooling (mean pooling in their case) is applied to obtain a single text embedding for each image, which is then used for similarity computation. Such trick increase GPU memory consumption because 1203 categories are expanded to $1203 \times 80 = 96,240$ text embeddings when using full templates.

Another trick, chunk dataset, not mentioned in their paper but present in their code, follows GLIP Li et al. (2022b) by splitting 40 categories into a single chunk with a total of $\lceil 1203/40 \rceil = 31$ chunks. Each image is duplicated 31 times, and each copy is associated with only 40 categories (3 categories for the last copy). During similarity computation, rare or long-tail categories receive relatively higher similarity scores because they are compared only within a small chunk rather than against all 1203 categories. This also reduces confusion when the chunk contains semantically close or ambiguous categories, making it easier for rare classes to stand out during similarity matching as shown in table7.

## A.6 DETAIL PARAMETERS

We demonstrate the detail parameters of OVRD-T, OVRD-L in table 8, including the training settings, model parameters and loss function.

