# OpenReview forum: "OVRD: Open-Vocabulary Relation DINO with Text-guided Salient Query Selection"
_ICLR.cc/2026/Conference — Submitted to ICLR 2026_

### Official Review · Reviewer_41M2 · 2025-10-28

**Soundness:** 3
**Presentation:** 3
**Contribution:** 3
**Rating:** 4
**Confidence:** 5

**Summary:**

This paper proposes Open Vocabulary Relation DINO (OVRD) with a text-guided salient query selection to choose image features most relevant to the text embeddings. OVRD captures the symmetric and fully-connected semantic relations with the aid of text-aware soft-mapping, and also models the directional and sparse relations to guide multimodal fusion and improve zero-shot detection performance.

**Strengths:**

Overall, the manuscript is well-organized and clearly written.

**Weaknesses:**

In the Introduction, the authors claim that recent methods [Cheng et al. (2024); Du et al. (2024); Wang et al. (2024)] insufficiently leverage semantic cues for guiding visual perception, and thus propose a text-guided mechanism as an enhancement. However, this argument lacks clarity and persuasiveness, as the cited works themselves fundamentally rely on text-guided or vision-language fusion strategies. The authors should provide a more precise and technically meaningful differentiation. Specifically, it is essential to clarify: What is the key limitation shared by the fusion strategies in [Cheng et al. (2024); Duetal. (2024); Wang et al. (2024)] that the proposed method overcomes?

The mathematical symbols in the manuscript need rigorous typesetting to enhance clarity and avoid ambiguity. It is common to use different fonts to distinguish scalars, vectors, and matrices. For instance, in the expression “I∈R^(H×W×3)”, I represents a matrix, while H and W are scalar dimensions.

Please provide visualizations to interpret the “Text-guided Salient Query Selection” module. Specifically, how do the selected salient regions correlate with the input text prompts? Can you also show a case where this process correctly identifies a novel object that a baseline method misses? Such evidence is key to validating the module's contribution.

**Questions:**

The mathematical symbols in the manuscript need rigorous typesetting to enhance clarity and avoid ambiguity. It is common to use different fonts to distinguish scalars, vectors, and matrices. For instance, in the expression “I∈R^(H×W×3)”, I represents a matrix, while H and W are scalar dimensions.

The pipeline in Figure 2 (b) raises two questions: What is the source of the “Points” and “Masks”? Are they generated by an external model or an upstream process?  There is logical redundancy: if the exact mask has already been obtained, the bounding box can be directly obtained. Why do we need a separate branch for bounding box detection? Please clarify the role of masks in this framework.

In Table 1, the results of metric APr and APc are not analyzed in detail. Additionally, the values of the APr and APc metrics for rare and common categories are significantly lower than those of SOTA methods on both the LVISMiniVal and LVISVal datasets. The paper claims strength in semantic relation modeling in open-vocabulary scenarios. However, the results show severely low APr and APc. Without convincing analysis and improved performance, the core claim of semantic relation modeling is unsubstantiated.

---

> ### Author Response · Authors · 2025-11-25
>
> We sincerely thank reviewer 41M2 for the insightful feedback and helpful suggestions. We address each concern point-by-point below.
>
> **Weakness1:**
>
> We concretely explain why YOLO-World Cheng et al. (2024), LAMI-DETR Duetal. (2024) and OV-DINO Wang et al. (2024) are insufficient in leveraging semantic cues for guiding visual perception.
>
> (1)Although YOLO-World introduces text-guided modulation, its fusion design is intentionally lightweight: a single global text embedding reweights pixel-level visual features with a sigmoid gating mechanism. This avoids the computational cost of cross-attention but also limits the depth and granularity of multimodal interaction, preventing sufficient exploitation of fine-grained semantic cues from the text.
>
> (2) LAMI-DETR performs “Language Embedding Fusion’’ by concatenating query features with text embeddings, but does not perform salient feature. Without selecting text-relevant tokens, all visual tokens participate in fusion, which makes the alignment weak and easily dominated by irrelevant regions.
>
> (3) While OV-DINO selects salient feature tokens, it does not select the corresponding salient reference points nor the valid masks. Some “salient’’ tokens may lie in padded regions after image resizing (no salient mask), and salient tokens receive no positional embeddings, making them spatially anonymous and forcing the decoder to rely purely on semantic similarity, which harms small-object and dense-scene detection.
>
> Our OVRD framework introduces a complete and spatially consistent TSQS design that simultaneously selects salient feature tokens, their aligned salient reference points and salient masks.
>
> We equip salient reference points with Vision RoPE, enabling the model to recover relative spatial relationships that are otherwise lost in flattened encoder features. Additionally, our Semantic Relation Self-Attention (SRSA) module further enhances multimodal fusion by explicitly modeling semantic dependencies among queries, enabling the detector to better leverage text-guided semantic cues.
>
> **Weakness2:**
>
> Thank you for the suggestion regarding mathematical notation.
>
> We agree that clear distinctions between scalars, vectors, and matrices are important for readability.
>
> In our manuscript, the expression I∈R^{H×W×3} follows the _standard notation widely adopted in computer vision literature_, where:
>
> - I denotes an image tensor (not a mathematical matrix),
> - H and W are scalar height and width,
> - the last dimension "3" indicates RGB channels.
>
> This notation is commonly used in foundational works such as **ViT**[1], **YOLO-World**, **YOLOE**, and many recent CV papers.
>
> To avoid ambiguity, we have additionally clarified in Section 3.1 that H and W represent the image resolution.
>
> **Weakness 3:**
>
> We fully agree that visual evidence is important for interpreting the proposed **Text-guided Salient Query Selection (TSQS)** module.
>
> To address this, we have **added extensive visualizations in the revised manuscript**:
>
> In **Appendix A.4**, we provide Visualization of salient tokens over text-guided relevance Heatmap, where:
>
> - We first construct a relevance heatmap between encoder tokens and text embeddings.
> - We then overlay the **selected salient tokens** on top of this heatmap.
> - The results clearly show that **TSQS consistently selects tokens in the most text-relevant regions**, and the density of these selected tokens increases around strongly semantic areas.
>
> This directly demonstrates the correlation between the selected salient tokens and the text prompts.
>
> Following the reviewer's request, we additionally provide a **side-by-side comparison with OV-DINO** in **Appendix A.3**, and our OVRD successfully detect small objects and densely crowded scenes.
>
> **Question1:**
> As described in Weakness2, this notation is commonly used in foundational works such as **ViT**, **YOLO-World**, **YOLOE**, and we directly use it.
>
> [1] A. Dosovitskiy, L. Beyer, A. Kolesnikov, D. Weissenborn, X. Zhai, T. Unterthiner, M. Dehghani, M. Minderer, G. Heigold, S. Gelly, J. Uszkoreit, and N. Houlsby, “An image is worth 16x16 words: Transformers for image recognition at scale,” ICLR, 2021.

---

> > ### Author Response · Authors · 2025-11-25
> >
> > **Question 2:**
> >
> > The "Points'' and "Masks'' in Figure 2(b) are not segmentation masks nor outputs from any external model; they are internal components generated directly from the encoder's multi-scale image features.
> >
> > As explained in Section 3.1, the _mask_ is simply the validity mask derived from the flattened multi-scale image feature, used to indicate padded positions and suppress attention in invalid regions-rather than providing any semantic or spatial extent of objects. Similarly, the _points_ are reference points produced from encoder features following the standard DETR pipeline, and are only used to construct positional embeddings. Neither the masks nor the points contain object-level spatial information, and therefore they cannot replace the bounding box regression branch. The detector still requires a dedicated box regression module to localize objects, refine coordinates, and align image regions with text embeddings. Thus, there is no redundancy: masks ensure valid attention regions, points supply positional cues, and the detection head remains responsible for generating object boxes.
> >
> > **Question 3:**
> >
> > A similar concern was mentioned by Reviewer 1gDH. We add analysis of  APr and APc in Section 4.3.
> >
> > We acknowledge that OVRD-L has lower APr and APc than OV-DINO, but it consistently achieves higher overall AP and APf. More importantly, OVRD-T shows strong improvements: with only 12 training epochs, it exceeds OV-DINO (24 epochs) by +6.9 AP, +15.1 APr, +9.6 APc, and +3.1 APf.
> >
> > In a controlled ablation using OV-DINO within our framework, our method further improves by +4.2 AP, +4.3 APr, +4.6 APc, and +3.6 APf. These consistent gains demonstrate that our semantic modeling provides substantial benefits.

---

### Official Review · Reviewer_1gDH · 2025-10-30

**Soundness:** 1
**Presentation:** 3
**Contribution:** 2
**Rating:** 2
**Confidence:** 4

**Summary:**

This paper introduces OVRD, which improves open-vocabulary object detection performance by utilizing semantic cues from text embeddings to guide the model's visual perception.

**Strengths:**

**Originality**:
- This paper's motivation is reasonable. Open-vocabulary object detection indeed needs to utilize cues from the text.

**Clarity**:
- This paper clearly explains its motivation and methodology.

**Weaknesses:**

- The experimental results in this paper fail to verify that the proposed method can enhance visual perception by leveraging text information: (1) In Table 1, the proposed method OVRD does not demonstrate consistently superior performance over existing methods, particularly on the LVIS rare categories. (2) The paper's ablation study only compares AP metrics under different settings but lacks a specific analysis of using the information from the text embeddings. This fails to show a direct link between the changes in AP values and the utilization of text information. (3) The paper does not provide any visualization results or examples. (4) The experiments are only conducted on LVIS benchmark. Other benchmarks such as COCO-OVD should be included.
- The paper's assertion that 'existing methods are insufficient in utilizing semantic cues from the text embeddings'  is insufficiently supported and lacks corresponding evidence. This is the paper's fundamental motivation. so it requires justification. I hope the authors can supplement this claim with more evidence, including experimental results, examples, or supporting references.

**Questions:**

- The main experimental results in Table 1 are under the zero-shot setting, but open-vocabulary object detection is different from zero-shot object detection. Given that the title includes 'open-vocabulary', could the authors please explain why the zero-shot setting was used?

---

> ### Author Response · Authors · 2025-11-25
>
> We sincerely thank reviewer 1gDH for the insightful feedback and helpful suggestions. We address each concern point-by-point below.
>
> **Weakness1:**
>
> We acknowledge that OVRD-L obtains lower APr and APc than OV-DINO on LVIS MiniVal and LVIS Val. However, our model consistently achieves higher AP and APf , demonstrating stronger overall detection capability. Importantly, our OVRD-T shows substantial improvements. When trained for only 12 epochs, it outperforms OV-DINO trained for 24 epochs on the same dataset by **+6.9 AP**, **+15.1 APr**, **+9.6 APc**, and **+3.1 APf**.
>
> Furthermore, in our ablation study where we adopt OV-DINO _within our unified framework_ as the baseline to ensure a fully controlled comparison, our method brings consistent and significant gains: **+4.2 AP**, **+4.3 APr**, **+4.6 APc**, and **+3.6 APf**. These results clearly demonstrate that our proposed components deliver meaningful improvements across all category groups - especially for rare and common classes, where semantic cues are most critical - and therefore substantiate the effectiveness of our semantic modeling approach
>
> **Weakness2:**
>
> Our base detector is DINO, and both OV-DINO and Grounding-DINO are also built upon DINO. The primary differences across these models lie in **how text embeddings are processed and utilized**. Therefore, changes in AP directly reflect **the effectiveness of leveraging textual information**, rather than modifications to the underlying detection architecture.
>
> Our ablation indeed reflects the effect of utilizing text embeddings, because the _only_ difference between settings is **how textual information interacts with visual queries**. All modules we ablate are directly conditioned on text embeddings:
>
> - **TSQS** selects salient feature tokens, reference points and masks _based on their similarity to text embeddings_.
> - RoPE-based positional embeddings are computed **only for the salient reference points selected by TSQS**.
> - **Semantic Relation Self-Attention** uses semantic relation distribution as attention weights (Eq. 7).
>
> Thus, AP changes **directly reflect** how effectively the model utilizes textual information, because all ablated components _are mechanisms that transform text embeddings into query selection and semantic relations_.
>
> These results confirm that performance gains stem specifically from improved **text-conditioned guidance**, rather than unrelated architectural changes.
>
> **Weakness 3:**
>
> We provide the visualization results in Appendix A.3 (A.4 in the original submission), and we explicitly mention at the end of Section 4.3 that these visualizations are included in the appendix.
>
> Following reviewer 41M2's suggestion, we additionally visualize the raw encoder features, the selected top-k salient tokens after our TSQS module, as well as a direct comparison against the visualizations from OV-DINO.
>
> These supplementary results further demonstrate that our method offers clear advantages in complex and densely crowded scenes, where the selected salient regions align better with text-relevant areas and produce more reliable detections.
>
> **Weakness 4:**
>
> COCO-OVD requires training on COCO, which is **incompatible with the zero-shot benchmark**. Despite this difference, we still provide **zero-shot COCO results** in Section 4.3 and Table 2. Our OVRD-L achieves the **best zero-shot COCO AP**, outperforming OV-DINO2 by **+0.6 AP**.

---

> > ### Author Response · Authors · 2025-11-25
> >
> > **Weakness5:**
> >
> > OV-DINO already points out that previous approaches suffer from: _"how to efficiently leverage the language-aware capability for region-level cross-modality fusion and alignment"_ (Abstract).
> >
> > Although OV-DINO significantly improves the text-guided query selection pipeline, several essential components remain missing, which lead to sub-optimal utilization of semantic cues and insufficient spatial reasoning:
> >
> > Our analysis shows that OV-DINO's text-guided query selection is incomplete: it selects only salient feature tokens but ignores their corresponding spatial reference points and valid masks, causing many selected tokens to fall into padded regions and preventing any spatial positional embedding from being preserved. As a result, OV-DINO's decoder receives semantically relevant but spatially anonymous tokens, which harms small-object detection and dense-scene reasoning.
> >
> > OVRD addresses these limitations with a **spatially consistent TSQS**, which jointly selects salient tokens, their aligned salient reference points, and valid masks, and further injects **relative positional cues via Vision RoPE** to restore spatial structure lost in OV-DINO. In addition, our **semantic relation self-attention (SRSA)** models inter-query semantic interactions and particularly improves rare-category recognition.
> >
> > Across experiments, OVRD consistently outperforms OV-DINO:
> >
> > - **Main results (Table 1)** show clear overall gains across settings.
> > - **Ablations (Table 3)** confirm the contribution of each component (+1.6 AP from spatial TSQS, +2.3 AP with RoPE, +4.3 AP with SRSA; SRSA alone gives +2.1 AP).
> > - **Visualizations (Appendix A.3)** further show that OVRD detects more objects-especially small and densely crowded ones-thanks to the restored spatial relations and stronger semantic cues.
> >
> > **Question 1:**
> > We follow the **standard evaluation protocol adopted by all recent open-vocabulary object detection methods**, including Grounding-DINO, OV-DINO, YOLO-World, YOLOE, Open-Det and report Fixed AP, Therefore, we adopt the same protocol **to ensure fair comparison** with prior work.

---

> > > ### Comment · Reviewer_1gDH · 2025-11-27
> > >
> > > The authors addressed most of my questions , but as reviewer Dm4g pointed out, this paper does not directly compare against the full version of OV-DINO, but rather against OV-DINO with the tricks removed. Although the authors provided a comparison between OV−DINO$^1$ and OVRD−T with the tricks included in Table 7 , I would like to see the comparison results for OV−DINO$^2$ and OVRD−L , as both of these models achieve higher accuracy.

---

> > > > ### Author Response · Authors · 2025-11-29
> > > >
> > > > Thank you for the suggestion. We have added the corresponding comparisons for OV-DINO$^2$ and OVRD-L. As shown in the updated Table 7, OVRD-L still surpasses OV-DINO2 by +0.8 AP, confirming the consistent advantage of our method. Full details are provided in the revised manuscript.

---

### Official Review · Reviewer_vZQg · 2025-10-31

**Soundness:** 2
**Presentation:** 3
**Contribution:** 2
**Rating:** 4
**Confidence:** 4

**Summary:**

This paper proposes OVRD, an open-vocabulary object detection method built upon DINO that aims to improve zero-shot detection performance. The main contributions include: (1) Text-guided Salient Query Selection that selects text-relevant image features along with their reference points and masks, (2) vision rotary positional embeddings for positional awareness enhancement, and (3) Semantic Relation Self-Attention that models sparse and directional semantic relations.

**Strengths:**

- The paper provides good intuition for modeling semantic relations in open-vocabulary scenarios, and the proposed components work together in a logical manner to enhance multi-modal fusion and text-guided visual perception.
- Comprehensive ablation studies. Table 2 demonstrates that each component contributes to the final performance, with detailed analysis of design choices.

**Weaknesses:**

- Limited technical novelty. The text-guided salient query selection is a straightforward extension of OV-DINO, by adding reference points/masks selection in Eq. (2). RoPE is borrowed from existing work. The semantic relation modeling, while interesting, only provides +1.4 AP gain as shown in table 2. The overall contribution feels like combining existing techniques.
- The re-evaluation of OV-DINO baselines described in section A.5 is problematic. The authors remove what they call "post-training tricks" which were actually design choices in the original OV-DINO method. This creates an unfair comparison where baseline are artificially lowered. The improvement over OV-DINO$^2$ is only 0.9 AP on LVIS Minival when comparing against the weakened baseline, making the true contribution unclear.

**Questions:**

N/A

---

> ### Author Response · Authors · 2025-11-25
>
> We sincerely thank reviewer vZQg for the insightful feedback and helpful suggestions. We address each concern point-by-point below.
>
> **Weakness1:**
>
> 1\. A similar concern was mentioned by Reviewer Dm4g.  Our TSQS module addresses essential components that were missing in both Grounding-DINO and OV-DINO, enabling a more complete and spatially consistent text-guided selection process.
>
> In OV-DINO, only salient feature tokens are selected, but the corresponding salient masks and salient reference points are not. These omissions lead to two critical issues:
>
> (1) after image resizing, many encoder locations fall into padded regions, and without selecting the valid mask, some “salient” tokens inevitably lie in invalid areas and introduce attention noise;
>
> (2) without selecting the aligned salient reference points, the salient memory tokens lack positional embeddings and therefore lose all spatial cues, causing the decoder to rely almost purely on semantic similarity—particularly harmful for small objects and dense scenes.
>
> We verify the importance of these missing components in Table 3: adding salient reference points and masks already improves OV-DINO by +1.6 AP, and using Vision RoPE further increases the gain to +2.3 AP.
> These results confirm that our TSQS design meaningfully completes the prior framework and yields tangible performance improvements.
>
>
> 2\. We acknowledge that RoPE is borrowed from existing work.
>
> However, **in our method, RoPE is not an independent trick**, but **a necessary component to fix a structural limitation introduced by TSQS**, and therefore forms an inseparable part of our overall design.
>
> Traditional absolute positional encodings, such as the sine positional embeddings used in DETR, encode the _absolute coordinates_ of each token. Yet after TSQS selects a sparse subset of salient tokens, **the original 2D spatial layout becomes disrupted**, making absolute positions meaningless. As a result, DETR-style positional encoding can no longer represent where these tokens lie relative to each other.
>
> To address this, **we adopt Vision RoPE to encode token-token relative geometry** information, which restores spatial awareness in this sparse selection scenario. This enables the model to retain fine-grained spatial relationships that are critical for small objects and densely crowded scenes. Thus, **RoPE in our framework is not merely borrowed-it is required to make TSQS spatially consistent and functional**. TSQS and RoPE together form a tightly coupled design, rather than two loosely attached modules.
>
> 3\. While the semantic-relation module shows a +1.4 AP improvement in Table 2, this number should not be interpreted as "only +1.4 AP" for several reasons:
>
> **(1) The gain is measured on top of an already strong TSQS + RoPE baseline.**
>
> Our semantic relation modeling is the _third_ stage in a progressive ablation.
>
> The baseline has already been strengthened by salient selections and positional enhancements.
>
> Obtaining an additional **+1.4 AP on top of these improvements is substantial**, because marginal gains naturally diminish as the baseline becomes stronger.
>
> We also evaluate only using semantic relation self-attention on baseline, and achieve +2.1AP improvement. Results and descriptions are shown in Table 3.
>
> **(2) Semantic relation brings meaningful gains, especially on rare categories**
>
> While the semantic relation module yields +1.4 AP overall, its impact is far from marginal.
>
> Crucially, it brings **a much larger improvement on rare categories (+3.1 AP_r)**, which are the most challenging. The improvement from our semantic-relation modeling is both **meaningful and impactful**, especially for long-tail recognition.
>
> **(3) In OVD benchmarks, even +1 AP is considered substantial.**
>
> Recent OVD papers (e.g., OV-DINO, YOLO-World) often emphasize improvements of **+0.5-1.0 AP** as meaningful due to the difficulty of zero-shot recognition. In OV-DINO, Later-LASF improves for 0.9 AP (18.3 to 19.2). In YOLO-World, RepVLPAN improves for 1.1 AP (22.4 to 23.5)

---

> ### Author Response · Authors · 2025-11-25
>
> 4\. Our framework is **not a collection of existing components**, but rather a **structurally necessary correction** to flaws in prior text-guided query selection designs (e.g., Grounding-DINO and OV-DINO).
>
> - **TSQS + RoPE is not borrowed-and-stacked.**
>
> Previous methods _select_ salient features but **fail to select the corresponding salient reference points and salient masks**, which leads to misaligned spatial positions, invalid tokens from padded regions, and severely degraded performance on small and dense objects.
>
> Our TSQS redesign _fixes this structural error_, and RoPE is required to restore **relative spatial geometry** after sparse selection, which cannot be solved by prior absolute positional encodings (e.g., sine PE).
>
> Thus, TSQS and RoPE are **two parts of one coherent mechanism**, not independent borrowed techniques.
>
> - **Semantic Relation Self-Attention introduces new capability.**
>
> This module explicitly models **semantic dependencies among queries**, enabling the model to better recognize rare and semantically ambiguous categories-long the main bottleneck in open-vocabulary detection.
>
> Therefore, the overall contribution is **not a mere combination** of existing components, but a **principled redesign** that corrects structural limitations of previous OVD frameworks and introduces **new semantic reasoning capability** that was previously missing.
>
> **Weakness2:**
>
> A similar concern was mentioned by Reviewer Dm4g.
>
> We provide a detailed explanation of these two techniques (multi-template pooling and chunked label sets) in Appendix A.5, including their implementation and empirical effects.
>
> Here we offer a concise clarification:
>
> - **Neither our method nor any baseline in the main comparison uses these tricks.** To keep the evaluation fair and consistent with prior open-vocabulary detectors, we follow the same setting and do not introduce extra techniques that were not used in previous works.
> - **To assess their impact independently**, we performed controlled ablations on OV-DINO1 and OVRD-T on LVIS MiniVal. As shown in Table 7, both tricks significantly boost AP even without any architectural changes, indicating that they benefit _all_ OVD models.
> - **Because these tricks help every method**, enabling them only for ours but not for other baselines would create an unfair and misleading comparison.
>
> Therefore, **all main results are reported without these auxiliary tricks** (except GLIP, following its original setting), ensuring a fair, aligned, and unbiased comparison across all methods.

---

### Official Review · Reviewer_Dm4g · 2025-11-01

**Soundness:** 2
**Presentation:** 2
**Contribution:** 2
**Rating:** 4
**Confidence:** 4

**Summary:**

This paper proposes OVRD, a zero-shot open-vocabulary detector built upon the DINO framework. It introduces three main components: a Text-guided Saliency Query Selection (TSQS) mechanism, Positional-Aware enhancements (RoPE), and a Semantic Relation Self-Attention (SRSA) module. The authors report improved results over OV-DINO and Grounding-DINO on the LVIS benchmark, with ablations showing the contribution of each component. While the core idea of using text to guide visual attention is clear, the work's novelty is limited. The technical contributions feel more like an integration and fine-tuning of existing ideas from its predecessors rather than a significant methodological leap.

**Strengths:**

1、The paper correctly targets a central challenge in OVD: how to better leverage text priors to guide the model's query selection and visual attention.
2、The authors rightly point out that many models fuse multimodal information too late (i.e., only at the classification stage). The SRSA module is a sensible attempt to address this by modeling semantic relationships earlier in the self-attention layers.
3、The method is built directly upon the popular DINO/DETR family, making it easy for the community to understand, reproduce, and compare against.

**Weaknesses:**

1、The novelty of the TSQS module is questionable. Language-guided query selection is already a core component of both Grounding-DINO and OV-DINO, making this contribution feel more like a minor, incremental improvement.

2、The core idea of injecting a relation matrix into self-attention has been previously explored in works like Relation-DETR. While the specific implementation may be new, the underlying concept is not.

3、The reported performance gains are quite small (+0.9 AP in some cases). More importantly, the authors admit to re-evaluating OV-DINO by removing some of its original techniques (e.g., template ensembling). This breaks the experimental protocol and results in an unfair, apples-to-oranges comparison.

**Questions:**

1、The paper claims to select "text-relevant" queries, but the mechanism is opaque. It's not specified what the T_CLS module is (e.g., a linear layer, MLP?) or how it computes relevance from a simple feature norm.

2、The paper mentions using RoPE but fails to specify where it's integrated into the architecture. More importantly, it lacks a direct ablation study to isolate RoPE's specific contribution to performance.

3、he evaluation protocol is not clearly defined. The paper needs to explicitly state which version of the LVIS dataset is used and whether the standard "Fixed AP" metric is reported, making it difficult to verify if the setup aligns with key baselines.

4、The ablation study is flawed, as it bundles multiple changes together. It's impossible to disentangle the true contribution of the proposed modules from known confounding factors, such as simply using more queries or longer text inputs.

---

> ### Author Response · Authors · 2025-11-25
>
> We sincerely thank reviewer Dm4g for the insightful feedback and helpful suggestions. We address each concern point-by-point below.
>
> **Weakness1:**
>
> Our proposed TSQS design fills several missing but essential components in Grounding-DINO and OV-DINO, leading to a more complete and spatially consistent text-guided query selection mechanism.
>
> Specifically, **OV-DINO only selects salient feature tokens**, but **does not select the corresponding salient reference points nor the salient masks**.
>
> However, both components are crucial:
>
> - Due to image resizing, many encoder locations correspond to **invalid padded regions**. Without selecting the **salient mask**, a portion of the "salient" tokens will inevitably fall in padding areas, introducing noise into the attention computation. Without selecting the **salient mask**, a portion of the "salient" tokens will inevitably fall in padding areas, introducing noise into the attention computation.
> - The **salient reference points**, selected from the encoder's candidate boxes, provide the positional information needed to compute **positional embeddings for salient memory tokens**. Without proper positional encoding, these memory tokens lose all spatial cues and are matched to queries **only by semantic similarity**, which is known to degrade performance on **small objects and densely crowded scenes**.
>
> We validate the importance of these missing components in **Table 3**.
>
> Adding **salient reference points + salient masks** to the OV-DINO baseline already brings **+1.6 AP**, and replacing sine PE with **vision RoPE** yields a total improvement of **+2.3 AP**.
>
> These results clearly demonstrate that our spatially consistent TSQS completes the design gap in prior text-guided selection modules and provides substantial real performance gains.
>
>
> **Weakness2:**
>
> We would like to clarify that **injecting a relation matrix into self-attention is not the core idea of our method**.
>
> Instead, it is **one possible mechanism** to operationalize our main contribution-enhancing **semantic relational awareness among queries** in the open-vocabulary setting.
>
> Relation-DETR models **geometric relations** between queries across decoder layers to improve localization consistency.
>
> In contrast, **our method focuses on modeling semantic relations between queries**, which is fundamentally different in both purpose and effect:
>
> - Relation-DETR: encourages cross-layer **positional relation modeling**
> - Ours: enhances **semantically grounded interactions** among queries in an open-vocabulary context
>
> Moreover, our semantic relation modeling works **in tandem with TSQS**, which boosts each query's text-related semantics. Together, they provide a principled way to integrate high-level semantic cues into self-attention-something prior geometric-relation works do not address.
>
> Thus, the relational matrix we use is **an implementation choice**, while the key contribution lies in enabling **semantic relation modeling** tailored for open-vocabulary detection, which is fundamentally distinct from Relation-DETR's positional relation modeling.

---

> ### Author Response · Authors · 2025-11-25
>
> **Weakness 3:**
>
> 1\. Regarding the reviewer's concern about the performance gain on LVIS MiniVal:
>
> Although OVRD-L improves OV-DINO2 by **+0.9 AP** on LVIS MiniVal, we emphasize that **MiniVal is a small subset** and does not fully reflect generalization ability.
>
> On the full **LVIS Val** set-which is significantly larger and more challenging-our method achieves a **much more substantial improvement of +2.7 AP**, demonstrating the robustness and scalability of our design.
>
> In addition, when comparing OVRD-T with OV-DINO1 **under the same training data and the same (shorter) number of training epochs**, OVRD-T shows **consistently strong improvements**:
>
> - **+6.9 AP** on LVIS MiniVal
> - **+5.9 AP** on LVIS Val
>
> These gains clearly indicate that our method provides **meaningful and consistent benefits across model sizes, datasets, and evaluation splits**, rather than only marginal improvements in isolated cases.
>
> 2\. We provide a detailed discussion of these techniques in **Appendix A.5**, including their implementation details, the experimental settings, and their effects when applied to our method.
>
> Below we briefly summarize the key points relevant to the reviewer's concern:
>
> - **Neither our method nor any of the compared baselines use these two tricks** (multi-template pooling and chunked label sets). To ensure a fair comparison with all prior open-vocabulary detectors, **we intentionally do not use any additional techniques** beyond those adopted in previous works.
> - **To further study their impact**, we conducted controlled ablation experiments on **OV-DINO1 and OVRD-T** on LVIS MiniVal. As shown in Table 7, **both tricks significantly boost performance even without modifying the model architecture**, demonstrating that they provide general benefits to all open-vocabulary detectors.
> - **However, precisely because these tricks benefit all methods**, enabling them for our model-but not for other baselines-would make comparisons **unfair and misleading**.
>
> For this reason, **all main-table results report numbers (except GLIP) without these auxiliary techniques**, ensuring a strictly aligned and unbiased comparison.
>
> **Question1:**
>
> 1\. The reviewer's concern appears to arise from a misunderstanding of what our method actually _selects_.
>
> Our module is named _text-guided query selection_ because it follows the naming convention commonly used in prior DETR-based works (e.g., Grounding-DINO, OV-DINO), where "query selection" refers to the process of determining which information from the encoder is used for **initializing** queries.
>
> As we describe clearly in Section 2.1 and further in the rebuttal for Weakness #1,
>
> our module selects the following three components:
>
> **Salient feature tokens** - top-K text-relevant encoder features.
>
> **Salient reference points** - spatial anchors derived from the coordinates of the selected encoder features (not from boxes).
>
> **Salient masks** - masks that remove padded regions so that selected features never fall into invalid areas after image resizing.
>
> 2\. Our _contrastive classifier_ (also referred to as the _contrastive head_) computes similarity between text embeddings and visual features。
>
> This module first **projects the text embeddings** into the same dimensionality as the visual features. Then, it computes the **cosine similarity** between every query feature and every projected text embedding. This cosine similarity (implemented as normalized dot product) directly forms the **contrastive logits** used for open-vocabulary classification.
>
> We have also **added a detailed explanation of this module in Section 3.1** of the revised paper, explicitly describing:
>
> - the text projection layer,
> - the L2-normalization of both modalities, and
> - the contrastive similarity computation.
>
> This clarifies that our classifier is not an additional network head but a contrastive module that directly measures alignment between visual queries and text embeddings.

---

> > ### Author Response · Authors · 2025-11-25
> >
> > Question2：
> >
> > **We explicitly use RoPE when computing positional embeddings for both salient reference points and queries** inside our _Salient Multi-Head Attention_ module.
> >
> > This is stated clearly in **Section 3.3 "Vision Rotary Positional Embeddings"**, where we describe how RoPE is injected to encode the spatial geometry of selected salient tokens and their corresponding reference points.
> >
> > To verify the necessity of positional embeddings and to compare different implementations, we conducted comprehensive ablation studies in **Table 5 ("Ablations on Positional Embeddings")**.
> >
> > The experiments include:
> >
> > - **No positional embedding**
> > - **Sinusoidal positional embedding**
> > - **Learnable positional embedding**
> > - **Rotary positional embedding**
> >
> > The results consistently show that **RoPE provides the best performance**, demonstrating that injecting spatially-aware rotary embeddings is crucial for improving alignment between salient visual tokens, reference-point geometry, and decoder queries.
> >
> > These findings justify our design choice and confirm that RoPE is the most effective positional encoding method in our architecture.
> >
> > Question 3:
> >
> > We would like to clarify that **we use the LVIS v1 dataset**, fully aligned with prior methods such as **YOLO-World, YOLOE, and OV-DINO**.
> >
> > In addition, the **Fixed-AP evaluation protocol** strictly follows the same settings as these methods which is stated in Section 4.1, ensuring that all comparisons are performed under a consistent and fair evaluation setting.
> >
> > Question4:
> >
> > **All experimental settings-including the number of queries, decoder depth, text encoder, and training schedule-are strictly kept identical to our baseline.**
> >
> > The exact configurations are listed in **Appendix Table 8** to ensure full transparency and reproducibility.
> >
> > To further isolate the contribution of each component, **Table 3 includes a dedicated ablation where we add only the proposed Semantic Relation Self-Attention on top of the baseline**, without introducing any additional changes. This directly reflects the module's standalone effect.
> >
> > Regarding RoPE, it **cannot be independently ablated** because RoPE is applied **specifically on the salient reference points selected by TSQS**, and these reference points do not exist without TSQS.
> >
> > Therefore, isolating RoPE in the absence of TSQS would not form a valid model variant and would not reflect the actual design intention of our positional encoding mechanism.

---

### Meta-Review · Area_Chair_4FyH · 2025-12-08

**Summary:**

The paper was evaluated by four reviewers. One recommended rejection, and the other three rated it below the acceptance threshold. Among the reviews, several concerns were raised, including limited novelty, incomplete or insufficient empirical evaluation, unclear explanation of the proposed method and network architecture, and questions regarding the fairness of the baseline comparisons.
Two reviewers also noted that the reported performance gains over the baselines were relatively modest. The authors responded to this point in their rebuttal and revised the paper accordingly.

After carefully considering the rebuttal and the updated paper, it is clear that the authors have made an effort to improve the paper. However, several of the weaknesses identified in the initial reviews remain insufficiently addressed, and the current version does not yet meet the standard required for acceptance.
I encourage the authors to improve the work, particularly by strengthening the empirical evaluation, clarifying the methodological contribution, and improving the presentation, and consider submitting to another venue.

**Reviewer Concerns:**

Two reviewers (Dm4g and 1gDH) pointed out that the reported performance gains over the baselines were relatively small, and the authors made an effort to address this concern in the rebuttal. However, some key issues that were raised by the reviewers remain unresolved. Including the clarity of the main motivation, the level of novelty and distinct contribution of the proposed model, the comprehensive empirical evaluation, and the lack of a clear and specified architectural design. These concerns were not sufficiently clarified in the rebuttal.

**Reviewer Scores:**

For reviewers Dm4g, 1gDH, and vZQg, some of their comments were addressed in the rebuttal, however, their main concern regarding the limited novelty of the contribution remains unresolved. It is therefore unlikely that any of these reviewers would have raised their scores had they been able to participate fully in the discussion.
Reviewer 41M2 raised several technical issues related to the experimental setup, visualisation results and the proposed architecture, most of which the authors addressed in their response. Therefore, this reviewer may increase their score slightly.

---

### Decision · Program_Chairs · 2026-01-26

Reject